## COMMENT

# Being prepared for emotionally demanding research

Amy Burrell[1✉], Benjamin Costello [1], William Hobson[2], Ralph Morton[3], Carolina Gutierrez Muñoz[2], Katie Thomas[4] & Juliane A. Kloess[5]

Research on topics such as child sexual abuse can be emotionally demanding for researchers in ways that surpass many other lines of work. Burrell et al. draw on their own experience to offer recommendations that may help increase researchers' resilience to these challenges.

## Main

Research in some domains, such as research into abuse or violence, can undeniably take a toll on researchers. But this research is important as it helps to explore challenging aspects of the world we live in and make meaningful recommendations for change. It is therefore essential for research teams to put measures in place to support researchers before and during working on these projects.

Emotionally demanding research (EDR)[1] is a label for "research that demands a tremendous amount of mental, emotional, or physical energy and potentially affects or depletes the researcher's health or well-being". EDR is not limited to research on sensitive issues (e.g., violence, abuse, mental health, chronic or terminal illness, and death). Research can also be emotionally demanding when it concerns topics similar to the personal/traumatic experiences of the researcher, when the researcher experiences traumatic life events while conducting a study, and also when unexpected events arise during research that were not previously identified as a sensitive issue[1].

However, despite all possible differences, the overarching experiences of researchers will often be similar. For example, hearing personal accounts of the ways in which certain life experiences have a lasting psychological and/or physical impact on people, and the way they engage with and relate to the world around them, can affect researchers regardless of the topic area[2]. This can be emotionally demanding for anyone, but is especially so for researchers who have to immerse themselves in their data and the material, and ultimately places them at risk of being negatively impacted by their work.

We are a team of researchers comprising a lead researcher, four research fellows, and two research assistants working across the disciplines of psychology, philosophy, and linguistics. The topic area we have been working on together for over 2 years is child sexual abuse, and we all have varying experiences of working on EDR projects in the past, including online child sexual exploitation and abuse, female genital mutilation, homicide, and robbery/burglary.

Based on our collective experiences, we reflect on what was helpful and worked well, and we make suggestions about what other research teams may wish to consider to improve researchers' resilience. We also offer practical recommendations (Box 1, Box 2). These are based on a number of strategies, which we developed over the years to help us manage and cope with working on EDR projects. On the basis of our collective experiences, we propose the following recommendations for consideration by research teams who work on EDR, broadly falling within the areas of: (i) support and wellbeing (Box 1), (ii) working patterns and environment, and (iii) knowing yourself and your limits (Box 2).

[1] School of Psychology, University of Birmingham, Edgbaston, UK. [2] Department of Psychology, University of Bath, Bath, UK. [3] Aston Institute for Forensic Linguistics, Aston University, Birmingham, UK. [4] Faculty of Engineering, University of Bristol, Bristol, UK. [5] School of Health in Social Science, The University of Edinburgh, Edinburgh, UK. ✉email: a.burrell@bham.ac.uk

---

**Box 1 | Recommendations**

**Support and wellbeing**

1.1 Provide advice and guidance to new team member(s) on the psychological impact that working with EDR may have on them. Ensure they understand what working with EDR involves and introduce them to other researchers who work with EDR, where possible.

1.2 Offer access to funded training opportunities and resources (e.g., https://www.svri.org/) that are relevant to the topic area of the EDR project (e.g., employer-led CPD, conference attendance, talks, workshops), as well as highlighting the importance of maintaining wellbeing for both career development and the researcher's health. Time for such opportunities should be provided within working hours.

1.3 Provide regular, independent psychological support for researchers to help them develop coping strategies, and to identify and manage potential triggers.

1.4 Arrange regular team meetings in order to give researchers access to supervision and encourage reflection and discussion among the research team.

1.5 Encourage the researcher(s) to establish strategies to promote healthy working relationships with other researchers/their colleagues, and work alongside one another when working on EDR, as well as engage in open conversations about what they may find difficult. Where a researcher does not work within a team or alongside other researchers, they should be provided with a space for reflection, beyond clinical supervision.

1.6 Foster a work environment where all members of the research team feel comfortable to discuss their personal experiences, without fear of judgement. Feeling able to ask for advice or support is vital.

1.7 Allow for and enable flexibility regarding the allocation of tasks. Where tasks cannot be redistributed, generous planning should facilitate taking breaks. In extreme cases, researchers may have to be supported to cease working on EDR.

---

**Box 2 | Recommendations (continued)**

**Working patterns and environment**

2.1 Organise suitable working arrangements that provide workspaces for researchers to work alongside one another for regular check-ins and support (e.g., to be able to express certain thoughts and feelings, and share them with someone who understands). Find appropriate alternatives for researchers who do not work within a team/alongside other researchers.

2.2 Discourage unhealthy working practices. This will be guided by individual circumstances and work patterns and may include avoiding reading potentially distressing material for extended periods, at night, or in private spaces.

2.3 Limit exposure to potentially distressing material by allocating researchers to other project tasks for at least 50% of their working week. Advocate for the importance of considering the mental health and wellbeing of researchers who work on EDR at an institutional level, and implementing relevant practices to support them (e.g., clinical supervision). This ought to form part of every academic institution's Code of Practice for Research.

**Knowing yourself and your limits**

3.1 Reflect and discuss your experiences among the research team, talk about what may be triggering or difficult to work on. Be open and transparent with colleagues about what you feel that you can and cannot work on. Sharing what you may be struggling with is a strength; it can help with processing thoughts and acts as a protective factor for your wellbeing.

3.2 Take regular breaks and encourage your colleagues to take a break.

3.3 Accept and respect if working with EDR is not/does not feel right for you. There are many reasons why you may not wish to embark on or join an EDR project, and there are just as many reasons for why you may wish to leave an EDR project. If working on EDR is not for you, it is by no means a reflection of you as a person or a professional.

---

## Why EDR can be challenging

**Potentially distressing material**. Even after several years' experience of working on EDR projects, researchers can still find the type of material they are exposed to difficult to manage. Our group has primarily been working with textual data involving descriptions of child sexual abuse. The content is explicit and of a potentially distressing nature, including detailed accounts of someone's actions, thoughts, feelings, or fantasies, as well as the abuse they have perpetrated or suffered at the hands of others. In this line of work, researchers may experience intrusive thoughts or images, which ultimately make the accounts more vivid. The material can also activate previous memories and has the potential to cause flashbacks. In addition, it is important to acknowledge here that a researcher's positionality (e.g., gender, race, ethnicity, sexual orientation, etc.) will undoubtedly play a role in their experience of working on certain types of EDR.

**Content that has personal meaning/relevance**. Working with potentially distressing material can be even more challenging when its topic area carries personal meaning or relevance. Some members of our group became parents in the project's early stages, with others having extended family members who are young children.

In this context, working with data that detail abusive interactions with children can be particularly shocking, and we found that having one's own children can make a huge difference to the way researchers process such content. In addition to empathising with the victims, it may trigger thoughts about one's own children going through something similar, causing an emotional, and, at times, visceral reaction, and disbelief at how anyone could harm a child in this way. The ways in which certain aspects of the material can cause a delayed reaction is surprising, and it can be incredibly distressing when these reactions occur during interactions with children.

For example, when a baby is distressed, a method for testing whether they might be hungry is to gently place the tip or part of the knuckle of one's little finger against their mouth. When one researcher did this one evening, and their baby began to suck on it, this brought back memories of the content they had worked on the previous week, where a baby had been abused by an adult. This immediately led to feelings of nausea and confusion, and questions about why this experience with their baby had caused their mind to revisit this particular memory at this moment in time. Understandably, this not only tarnished the commonplace action of using one's little finger to test whether their baby was hungry, but it also negatively affected their mood, and caused anxiety and confusion around how to interact with their baby in the future.

A common experience across the group of researchers was also being reminded of content that involved nappy fetishes when changing the nappy of a young child, or associating the everyday cries of a baby/child with those of a baby/child being abused, or being aware of how offenders interpret the normal actions of children (e.g., arching their back) as some form of sexual invitation.

**Being un(der)prepared**. The level of preparation offered by supervisors or line managers has the potential to have a

significant impact on one's ability to manage working with EDR. For researchers new to the area (especially in the context of student research projects), it is important to receive a warning about how working with EDR may affect them. This should be accompanied by appropriate support to develop strategies that might help to manage this. While it may not always be possible to provide this, particularly in cases where early access to the nature of the research is restricted for reasons of sensitivity or confidentiality, researchers should be informed about what to expect.

Even with a general sense of the kind of data one will encounter, it is difficult to know in advance which content may be difficult to read, process, or analyse, or what may be triggering at a later point in time. Sometimes it is a feeling of déjà vu, encountering an object or scene, hearing a particular sound, smelling a particular scent, or having a particular feeling, which can then act as a trigger to revisit particular content or material. In such cases, we found it especially helpful to be able to talk to and reflect together with our colleagues.

Given this unpredictability, we would suggest that it is important for supervisors and line managers to cost for regular psychological support when applying for research funding, and thoroughly prepare researchers prior to working with EDR, particularly those who have not worked on EDR projects before. This can be as simple as explaining the nature of the data (e.g., text, visual), but should also include discussions about potential reactions to material and how to access support (see Sprang et al.[3] for a discussion of how organisational efforts can improve an individual's perceived level of distress).

**Limiting, containing, and managing exposure**. It is essential to consider when and where to conduct work on EDR projects and we found this one of the most valuable factors for our mental health and wellbeing. At the beginning of the research project, we were advised to only work on data at one of the research institutions and not at home.

However, circumstances can force researchers to deviate from the recommended guidance. Research on our project was affected by the COVID-19 lockdowns and transport strikes; this is an example of how achieving a separation between work on EDR and other activities can become dependent on personal living arrangements and external factors. Some researchers can create workspaces in spare rooms or designated areas in their homes, but others are not able to designate a space solely for work purposes. This, in turn, sometimes results in unhealthy working practices, such as working in personal spaces (e.g., one's bedroom) and/or outside of one's usual work pattern (e.g., working late into the night), which can ultimately make it more difficult to decompress and switch off from the material.

Other research activities must be conducted on secure sites not always within a commutable distance from home, requiring overnight stays. On the one hand, working on a secure site helps with enforcing a clear separation between work and home life, and can make it easier to manage working on EDR. On the other hand, working on a secure site, and thus away from home, acts as a physical separation from one's support network, making it harder to transition from a difficult work day to a more 'normal' evening, and thereby reducing one's ability to seek support.

The importance of one's professional support network was highlighted to us during COVID-19, when we were not able to socialise or decompress together as a team; this limited check-ins with one another during working hours and affected our ability to provide informal support within the team. Taken together, such unforeseen adverse conditions can have a significant impact on levels of physical tiredness and emotional exhaustion.

## Managing EDR projects

Research projects of a sensitive nature entail a more complex arrangement of contracts, ethics approval, and security vetting of researchers than standard research projects, and supervisors and line managers of EDR projects should be aware of what these complexities mean. Managing an EDR project takes a substantial amount of time and can often encounter delays. Researchers starting on the project at different points in time can impact on training and progress, especially when it is key that all researchers are consistent in their approach to data collection, management, and analysis.

Supervision and training through online communication channels may not be appropriate for some EDR projects, which instead often require in-person meetings due to their sensitive and confidential nature, therefore further adding to travel and time away from home.

We believe it is vitally important that supervisors or line managers are (i) open and transparent about the difficulty of working with EDR, (ii) understanding that this impacts people in different ways, and (iii) flexible in terms of taking this into account when allocating tasks. Researchers' willingness to help one another is key to creating an environment within the team where people feel comfortable to approach a colleague to discuss any difficulties.

Lastly, EDR that is of a particularly sensitive and confidential nature limits what you can share with others, both in your personal and professional lives. This also comes with restrictions around data security, requiring careful thinking about handling, processing, and storing data, which can be anxiety-provoking and cause stress in light of the potential implications of a data breach. Data held by police and government agencies often do not lend themselves to academic research/scientific enquiry in terms of the format in which they are collected and stored, often requiring multiple rounds of redaction, processing, and reworking to make them usable, resulting in additional time being spent on EDR.

We think that supervisors or line managers must be aware of and accommodate for the fact that the complexities of EDR projects can cause further delays and impact their timeline in a way that is in conflict with stringent deadlines imposed by funders, which can cause overworking among researchers.

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

## Author contributions

All authors (A.B., B.C., W.H., R.M., C.G.M., K.T., and J.A.K.) contributed equally to this work. All authors drafted individual sections and collectively refined the sections into this paper. All authors reviewed and approved the paper prior to submission.

## Competing interests

The authors declare no competing interests.
