## [Peer Review File · Communications Psychology]

13th Jun 23

Dear Amy,

Your Comment titled "Working on emotionally demanding research" has now been seen by 2 referees, whose comments appear below. Both reviewers highlight the important contribution you make. In the light of their advice I am delighted to say that we are happy, in principle, to publish it in *Communications Psychology* under a Creative Commons 'CC BY' open access license without charge.

We will not send your revised paper for further review if, but we very strongly encourage you to make most of the referees' detailed and constructive feedback. If the revised paper is in *Communications Psychology* format, in accessible style and of appropriate length, we shall accept it for publication immediately.

You will find that the reviewers are in disagreement about the utility of the framework of "emotionally demanding research". Such disagreements between experts are not uncommon and because your piece is an opinion contribution, not a research article, there is no need to resolve the conflict. However, the constructive criticism offered by Reviewer #2 (including on the manuscript file itself), may help you to make the opinion piece appealing to a broader audience without necessarily changing the framing.

To aid your revisions, I attach both, the edited manuscript provided by the referee and a manuscript file that includes editorial comments. To limit confusion, I used the manuscript file provided by Reviewer #2 (which includes their comments) and added editorial guidance into that document. This way, you only have to work from one source. Please do not hesitate to reach out if any of these issues are unclear. Please note that you will need to download the attachment to Microsoft Word or another programme that can read .docx files to see the comments and tracked changes.

* *Communications Psychology* uses a transparent peer review system. On author request, confidential information and data can be removed from the published reviewer reports and rebuttal letters prior to publication. If you are concerned about the release of confidential data, please let us know specifically what information you would like to have removed. Please note that we cannot incorporate redactions for any other reasons.

*If you have not done so already, please alert me to any related manuscripts from your group that are under consideration or in press at other journals, or are being written up for submission to other journals (see www.nature.com/authors/editorial_policies/duplicate.html for details).

FORMATTING GUIDELINES:

You will find a complete list of formatting requirements following this link:

<https://www.nature.com/documents/commsj-style-formatting-checklist-comment.pdf>

Please use the checklist to prepare your manuscript for final submission. In the following, I also highlight some issues of particular importance.

** Title

Titles should be descriptive of the main message your manuscript conveys and should not exceed 90 characters (including spaces). Please note that punctuation is not allowed, nor are titles of the following format: "title: subtitle".

** Preface

The paper's preface (up to 40 words; without references) should serve both as a general introduction to the topic, and highlight your position or proposal. Because we hope that researchers across all fields of psychology will be interested in your work, the preface should be as accessible as possible.

** Length

At this stage, although the revision may require some expansion of the text, please try to limit additional text to 250 words or fewer in total.

** Main text

Please provide three or four section headings in the main text. These should relate to the content of the article rather than being generic. Headings should be no longer than 30 characters (including spaces) and should not use punctuation.

* References

References appear as superscript Arabic numerals, in order of mention. The reference list mentions references in the numerical order in which they are mentioned in the main text. If a reference is cited more than once, the same number is used throughout the text and the reference receives a single entry in the reference list.

Only papers that have been published or accepted by a named publication should be in the reference list (preprints and citations of datasets are also permitted). Unpublished/Submitted research should not be included in the reference list; it should only be mentioned briefly and parenthetically in the main text. Note that no major arguments should rely on unpublished research.

Published conference abstracts and URLs for web sites should be cited parenthetically in the text, not in the reference list.

Footnotes are not used.

* Competing interests

Please include a "Competing interests" statement after the References. Note that we ask authors to declare both financial and non-financial competing interests. For more details, see <https://www.nature.com/authors/policies/competing.html>. If you have no financial or non-financial competing interests, please state so: "The authors declare no competing interests."

SUBMISSION INFORMATION:

* If you wish, you may also submit a visually arresting image, together with a concise legend, for consideration as a 'Hero Image' on our homepage. The file should be 1400x400 pixels and should be uploaded as 'Related Manuscript File'. In addition to our home page, we may also use this image (with credit) in other journal-specific promotional material.

* Your paper will be accompanied by a two-sentence editor's summary, of between 250-300 characters, when it is published on our homepage. Could you please approve the draft summary below or provide us with a suitably edited version.

Research on topics such as child sexual abuse is emotionally demanding for researchers. Research groups embarking on these topics may benefit from considering a set of practical guidelines.

In order to accept your paper, we require the following:

* A cover letter describing your response to our editorial requests.

* A separate document summarising the changes you made in response to the referees (please include the referees' comments in this document).

* The final version of your text as a Word or TeX/LaTeX file, with any tables prepared using the Table menu in Word or the table environment in TeX/LaTeX and using the 'track changes' feature in Word.

At acceptance, the corresponding author will be required to complete an Open Access Licence to Publish on behalf of all authors, declare that all required third party permissions have been obtained.

Please note that your paper cannot be sent for typesetting to our production team until we have received this information; **therefore, please ensure that you have this ready when submitting the final version of your manuscript.**

ORCID

Communications Psychology is committed to improving transparency in authorship. As part of our efforts in this direction, we are now requesting that all authors identified as 'corresponding author' create and link their Open Researcher and Contributor Identifier (ORCID) with their account on the Manuscript Tracking System (MTS) prior to acceptance. ORCID helps the scientific community achieve unambiguous attribution of all scholarly contributions. For more information please visit <http://www.springernature.com/orcid>

For all corresponding authors listed on the manuscript, please follow the instructions in the link below to link your ORCID to your account on our MTS before submitting the final version of the manuscript. If you do not yet have an ORCID you will be able to create one in minutes.

IMPORTANT: All authors identified as 'corresponding author' on the manuscript must follow these instructions. Non-corresponding authors do not have to link their ORCIDs but are encouraged to do so. Please note that it will not be possible to add/modify ORCIDs at proof. Thus, if they wish to

have their ORCID added to the paper they must also follow the above procedure prior to acceptance.

To support ORCID's aims, we only allow a single ORCID identifier to be attached to one account. If you have any issues attaching an ORCID identifier to your MTS account, please contact the Platform Support Helpdesk.

[link redacted]

We hope to hear from you within two weeks; please let us know if the process may take longer.

Best wishes

Marike

Marike Schiffer, PhD
Chief Editor
Communications Psychology

REVIEWERS' COMMENTS:

Reviewer #1 (Remarks to the Author):

This comment addresses the challenges of working on research projects that may be emotionally demanding, and provides strategies for managing the toll such research may take on scholars. This is a useful contribution and likely to be of interest to other researchers. It is especially important for senior scholars/PIs to consider this topic and to address it with junior scholars, graduate students, etc., who may be new to research and may not feel comfortable speaking up if they are struggling with the content of a research project.

The paper is well written and accessible and could easily be assigned in undergraduate or graduate research courses as a supplement to discussions of challenges that can occur in the research process.

I have a couple of comments for the authors, which are offered as suggestions for expanding thinking on this subject. I defer to them on whether to address these items in this specific manuscript. First, it may be worth noting that the researchers' positionality (e.g., race/ethnicity, gender, sexual orientation, etc.) may lead some topics to be more emotionally demanding for some scholars than others even if they don't have direct personal experience (for example, images of police violence may be emotionally demanding for individuals from communities that suffer from systemic police brutality, even if they themselves haven't experienced it). Second, the practical recommendations generally refer to a research team; however, some researchers will be

working alone or nearly alone on EDR and may not have team members to debrief with, reach out to, etc. Addressing how individuals without a research team network to rely on can apply these in their situation could further assist those facing challenges with emotionally demanding research.

As a small editorial comment, item 2.4 and the first sentence of 3.1 seem to be repetitive and may be able to be combined or reworded.

Overall, I believe this is a useful contribution and support its publication.

Gwen Sharp

Reviewer #2 (Remarks to the Author):

This is a very important and relevant subject. I commend the authors on formulating a statement and bringing it to the fore. Their collected wisdom and collective experience demonstrate a close experience with this subject matter.

I have made my comments throughout the manuscript, which I have attached. My comments can be grouped in to three categories:

1. The label:« emotionally demanding research» is insufficient and misleading. The demands, as you state directly after this term, encompass the physical body, the unconscious (and I would also argue: social behaviour and relationships). Although you are building on the work of Kumar and Cavallero, a more generic and inclusive terms, such as "personally challenging research" is indicated.

2. Much of this content refers to concepts that are well developed in the trauma literature. These concepts need to be acknowledged. Using them more formally will also help the authors to better organize the text. It is not necessary that researchers are diagnosed with PTSD to have a sense of their world being shattered and meanings changed/alterd (such as the sucking baby). I have indicated what content is refereing to previously theorized content and where this needs to be indicated.

3. The organization of the piece is not tight. It needs to be revised for structure and voice.

I was very moved by the authors' respectful and strength-based approach, also the in-house strategies they describe are largely supported in the trauma literature! It was fascinating to see how they came to much of this content without the help of that literature.

Working on emotionally demanding research

Amy Burrell¹ (a.burrell@bham.ac.uk), Benjamin Costello¹ (b.d.costello@bham.ac.uk), William Hobson² (wghobson1@gmail.com), Ralph Morton³ (r.morton2@aston.ac.uk), Carolina Gutierrez Munoz² (cigm20@bath.ac.uk), Katie Thomas⁴ (ks22633@bristol.ac.uk), & Juliane A. Kloess⁵ (juliane.kloess@ed.ac.uk)

¹School of Psychology, University of Birmingham, Edgbaston, UK

²Department of Psychology, University of Bath, Bath, UK

³Aston Institute for Forensic Linguistics, Aston University, Birmingham, UK

⁴Faculty of Engineering, University of Bristol, Bristol, UK

⁵School of Health in Social Science, The University of Edinburgh, Edinburgh, UK

* All authors contributed equally to this paper

Preface

Research on topics such as child sexual abuse is emotionally demanding for researchers in ways that surpass many other lines of work. Burrell et al. draw on their own experience to offer recommendations for researchers embarking on these topics.

Main

Research in some domains, such as research into abuse or violence, can undeniably take a toll on researchers. But this research is important as it helps to explore challenging aspects of the world we live in and make meaningful recommendations for change. It is therefore essential for research teams to put measures in place to support researchers before and during working on these projects.

Emotionally demanding research (EDR) [1] is a label for “research that demands a tremendous amount of mental, emotional, or physical energy and potentially affects or depletes the researcher’s health or well-being”. EDR isn’t limited to research on sensitive issues (e.g., violence, abuse, mental health, chronic or terminal illness, and death). Research can also be emotionally demanding when it concerns topics similar to personal/traumatic experiences of the researcher, when the researcher experiences traumatic life events while conducting a study, and also when unexpected events arise during research that were not previously identified as a sensitive issue [1].

However, despite all possible differences, the overarching experiences of researchers will often align. For example, hearing personal accounts of the ways in which certain life experiences have a lasting psychological and/or physical impact on people, and the way they engage with and relate to the world around them, can affect researchers regardless of topic area [2]. Even though a researcher might have no personal experience of the phenomenon they are researching, or hold different views to their research participants, there is often some underlying element that makes research narratives resonate or personally relevant, and can therefore be accompanied by certain feelings or emotions (because, for instance, it is relatable or jarring). This can be emotionally demanding for anyone, but is especially so for researchers who have to immerse themselves in their data and the material, and ultimately places them at risk of being negatively impacted by their work.

We are a team of researchers comprising a lead researcher, four research fellows, and two research assistants working across the disciplines of psychology, philosophy, and linguistics. The topic area we

Commented [A1]: Reviewer #2: I find the label « emotionally demanding » insufficient and misleading. The demands, as you state directly after this term, encompass the physical body, the unconscious (and I would also argue: social behaviour and relationships). Although you are building on the work of Kumar and Cavallero, a more generic and inclusive terms, such as “personally challenging research” is indicated.

Commented [A2]: Editor: The reviewer here disagrees with your framing. You are not obliged to change the manuscript as they instruct, but you are very welcome to incorporate their advice and change the paragraph accordingly if you wish. In this case, you will need to change the title, too. Note that Reviewer #1 had no concerns about the EDR framing, so there is no pressing need to change it.

Commented [A3]: Editor: Adding 1 citation here makes sense, but bear in mind that this is an opinion piece and exhaustive citations are not necessary.

Commented [A4]: Reviewer #2: This is called shared trauma, or shared traumatic experiences. The work of germinal shared trauma researchers should be cited here.

Commented [A5]: Reviewer #2: What does this mean?

Commented [A6]: Reviewer #2: I thought that you already established 2 pathways: 1. Sensitive issues and 2. Shared trauma. You might consult the SAMHSA definition of trauma (the three E’s) that explains how things that impact su are related to our experience of them- which is entirely subjective.

Commented [A7]: Editor: Some clarification here in response to the reviewer’s comments would be helpful, but note that you do not need to make the work more academic by using the SAMHSA definition if you don’t find it resonates with what you want to say.

have been working on together for over ~~two~~2 years is child sexual abuse, and we all have varying experiences of working on EDR projects in the past.

Based on our collective experiences, we reflect on what was helpful and worked well, and we make suggestions around what other research teams may wish to consider.

Why EDR can be challenging

Potentially distressing material

Even after several years' experience of working with EDR projects, researchers can still find the type of material they are exposed to difficult to manage. Our group has primarily been working with textual data involving descriptions of child sexual abuse. The content is explicit and of a potentially distressing nature, including detailed accounts of someone's actions, thoughts, feelings, or fantasies, as well as the abuse they have perpetrated or suffered at the hands of others. In this line of work, researchers may experience intrusive thoughts or images, which ultimately make the accounts more vivid. The material can also activate previous memories and has the potential to cause flashbacks.

Content that has personal meaning/relevance

Working with potentially distressing material can be even more challenging when its topic area carries personal meaning or relevance. Some members of our group became parents in the project's early stages, with others having extended family members who are young children.

In this context, working with data that details abusive interactions with children can be particularly shocking, and we found that having one's own children can make a huge difference to the way researchers process the content related to children. In addition to empathising with the victims, it may trigger thoughts about one's own children going through something similar, causing an emotional, and, at times, visceral reaction, and disbelief at how anyone could harm a child in a particular way. The ways in which certain aspects of the material cause a delayed reaction can be surprising, and it is incredibly distressing when these reactions occur during interactions with children.

For example, when a baby is distressed, a method for testing whether they might be hungry is to gently place the tip or part of the knuckle of one's little finger against their mouth. When one researcher did just this one evening, and their baby began to suck on it, this brought back memories of the content they had worked on the previous week, where a baby had been orally abused by an adult. This immediately led to feelings of nausea and confusion, and questions around why this experience with their baby had caused their mind to revisit this particular memory at this moment in time. Understandably, this not only tarnished the commonplace action of using one's little finger to test whether their baby was hungry, but it also negatively affected their mood, and caused anxiety and confusion around how to interact with their baby in the future.

A common experience across the group of researchers was also being reminded of content that involved nappy fetishes when changing the nappy of a young child, or associating the everyday cries of a baby/child with those of a baby/child being abused, or being aware of how offenders interpret the normal actions of children (e.g., arching their back) as some form of sexual invitation.

Being un(der)prepared

The level of preparation offered by supervisors or line managers has the potential to have a significant impact on one's ability to manage working with EDR. For researchers new to the area (and especially in the context of student research projects), we think it is important to receive a prior

Commented [A8]: Editor: Reviewer #1 said: "First, it may be worth noting that the researchers' positionality (e.g., race/ethnicity, gender, sexual orientation, etc.) may lead some topics to be more emotionally demanding for some scholars than others even if they don't have direct personal experience (for example, images of police violence may be emotionally demanding for individuals from communities that suffer from systemic police brutality, even if they themselves haven't experienced it)."

If you agree, if this dovetails with your experience, I would encourage you to include the point somewhere in this section.

Commented [A9]: Reviewer #2: This is a very brief description, unbalanced with the next section- it is unclear how this is different without corresponding examples. You might draw on research describing mirror neuron activity in experiencing what another person experiences or in the transference/countertransference literature regarding how the professional receives and digests the material.

Commented [A10]: Editor: We editorially advise against citing more research here – it's not needed in an opinion paper and references to mirror neurons wouldn't add value.

Commented [A11]: Reviewer #2: Again, this is all well described in constructivist self-development theory (McCann and Pearlman)

Commented [A12]: Editor: You may or may not incorporate this reference. You do not need to expand on the text, as you're referencing personal experience.

Commented [A13]: Reviewer #2: CSDT again, this is about changes to the way one makes meaning and interprets their world.

Commented [A14]: Editor: No need to respond.

warning about how working with EDR may have an impact and to be supported to develop strategies that might be helpful to manage this. This may not always be possible, particularly in cases where prior access to the nature of the research is restricted for reasons of sensitivity or confidentiality. However, where it is possible, prior warning should be provided.

Commented [A15]: Editor:Rephrasing may be helpful to aid readability.

Commented [A16]: Reviewer #2: Too long, break up

Even with a general sense of the kind of data one will encounter, it is difficult to know in advance which content may be difficult to read, process, or analyse, or what may be triggering at a later point in time. Sometimes it is a feeling of déjà vu, encountering an object or scene, hearing a particular sound, smelling a particular scent, or having a particular feeling, which can then act as a trigger to revisit particular content or material.

Given this unpredictability, it is important that supervisors and line managers thoroughly prepare researchers for working with EDR, particularly those who have not worked on EDR before. This can be as simple as explaining the nature of the data (e.g., text, visual), but should also include discussions about potential reactions to material and how to access support.

Limiting, Practical-circumstancescontaining and managing exposure.

It is important to consider when and where to conduct work on EDR projects and we found this one of the most important factors for our mental health and wellbeing. At the beginning of the research project, we were advised to only work on data at one of the research institutions and not at home.

Commented [A17]: Reviewer #2: What you are describing here is the potential for post-traumatic stress, distress, and challenges to meaning-making. Warning people and sensitizing them is not sufficient. Now you need to acknowledge Trauma-Informed care literature and perhaps review the recommendations of Sprang for how to be a secondary traumatic stress informed « organization ». Warning people that they might be about to receive traumatic material is not sufficient to protect them against it- it is somewhat preventative in that, at least they will not feel odd or alone, but how will they/you then manage it?

Circumstances can force researchers to deviate from the recommended guidance. Research on our project was affected by the COVID-19 lockdowns and transport strikes; achieving a separation between work on EDR and other activities then becomes dependent on personal living arrangements. Some researchers can create work spaces in spare rooms or designated areas in their homes but others are not able to designate a space solely for work purposes. This, in turn, sometimes results in unhealthy working practices, such as working in one's bedroom and working late into the day.

Commented [A18]: Editor:This Comment piece is about what you learned from your experience and you don't need literature to validate the insight you gained. However, if you think it's useful, you can of course add a sentence saying that there is literature on how to prepare, or similar, and add a single reference.

Commented [A19]: Editor: Ref#2 changed the subheading, you don't have to adopt this.

Other research activities have to be conducted on secure sites not always within a commutable distance from home, requiring overnight stays. On the one hand, working on a secure site helps with enforcing a clear separation between work and home life, and can make it easier to manage working on potentially distressing material. On the other hand, working on a secure site acts as a physical separation from one's support network, making it harder to transition from a difficult work day to more 'normal' everyday life.

Commented [A20]: Reviewer #2: Judgmental. Make the case for why those practices are unhealthy if you believe them to be- because of this job in particular ? All work? Reformulate.

The importance of one's professional support network was highlighted during COVID-19 when we were not able to socialise or decompress together as a team; this limited check-ins with one another during working hours and impacted on our ability to provide informal support within the team. Taken together, this can have a significant impact on levels of physical tiredness, and emotional exhaustion.

Commented [A21]: Editor: No need to respond unless you think it's helpful.

Commented [A22]: Editor: This doesn't need an example, but perhaps a half-sentence or sentence to explain it a bit better.

Commented [A23]: Reviewer #2: Unclear. Needs an example.

Commented [A24]: Reviewer #2: These are important strategies in line with what the trauma literature tells us is effective to manage exposure...

Commented [A25]: Editor: No need to respond.

Managing EDR projects

Research projects of a sensitive nature entail a more complex arrangement of contracts, ethics approval, and security vetting of researchers than standard research projects, and supervisors and line managers of EDR projects should be aware of what these complexities mean. Managing an EDR project takes a substantial amount of time and can often encounter delays. Researchers starting on the project at different points in time can impact on training and progress, especially when it is key that all researchers are consistent in their approach to data collection, management, and analysis.

Commented [A26]: Editor: No need to respond

Commented [A27]: Reviewer #2: That is true of much research

Supervision and training through online communication channels may not always be appropriate for EDR, which instead often requires in-person meetings, therefore further adding to travel and time away from home.

Commented [A28]: Reviewer #2: Explain

Commented [A29]: Editor: It may indeed be helpful to explain why this is the case

We believe it is vitally important that supervisors or line managers are (i) open and transparent about the difficulty of working with potentially distressing material, (ii) understanding that this impacts on people in different ways, and (iii) flexible in terms of taking this into account when allocating different tasks. Researchers' willingness to help one another when one of them found a particular case or type of material difficult is key to creating an environment within the team where people feel comfortable to approach a colleague to discuss any difficulties.

Commented [A30]: Editor: Rather than responding to the referee in this paragraph, it would make sense to review the degree to which these issues are addressed in your list of recommendations and potentially revise these.

Lastly, the restrictions around confidentiality and data security impact on who you can talk to about what you do, and require careful thinking about the process of sharing information, as well as distributing files among the research team. This can undoubtedly be anxiety-provoking and cause stress. Data held by police and government agencies often do not lend themselves to academic research/scientific enquiry in terms of the format they are collected and stored in, resulting in additional time being spent on the data to make it usable.

Commented [A31]: Reviewer #2: This needs to be expanded, see it through to the end, i.e., what if someone is REALLY struggling- what should the supervisor do then? What limits (if any) should there be around personal content, etc.

We think that supervisors or line managers must be aware and accommodate for the fact that the complexities of EDR can cause further delays and impact a project's timeline in a way that is in conflict with stringent deadlines imposed by funders, which can cause over-working in the team.

Commented [A32]: Editor: A bit more explanation/an example may indeed be helpful.

Commented [A33]: Reviewer #2: I find this confusing, please make a clear link with the toll this takes on researchers through an example.

Practical recommendations

Over the years, we have developed a number of strategies to help us manage and cope with working on EDR. On the basis of our collective experiences, we propose the following recommendations for consideration by research teams who work on EDR, broadly falling within the areas of: (i) support, (ii) working patterns and environment, and (iii) knowing yourself and your limits.

Commented [A34]: Editor: Maybe it would be better to say what this means for what the supervisor has to do?

Commented [A35]: Reviewer #2: This is in a section called « managing » yet it is simply a statement of the challenges...

1. Support

- 1.1. Provide advice and guidance to researchers/new team members on the psychological impact that working with EDR may have on them. Ensure they understand what working with EDR involves, and introduce them to other researchers who work with EDR, where possible.
- 1.2. Offer access to training and resources such as? that are relevant to the topic area of the EDR, as well as highlighting the importance of maintaining wellbeing for both career development and the researcher's health. Is that what the resources are for? Are they free? Is time using them remunerated? Is this another challenge of "EDR"?
- 1.3. Encourage researchers to establish positive working relationships [establish strategies to promote healthy working relationships] with their colleagues, and work alongside one another when working on difficult material, as well as engage in open conversations about what they may find difficult. One of the aspects we found most helpful was being able to discuss the EDR and topic area with our colleagues. I suspect that this was with colleagues who had specific skills and characteristics (Mentalizers)... not everyone will be good at this.
- 1.4. Foster a work environment where all members of the research team feel comfortable to discuss their personal experiences, without fear of judgement. Feeling able to ask for advice or support is vital, and sharing what you may be struggling with is not a weakness; strength; it can help process your thoughts and act as a protective factor against any impact on your wellbeing in the future.
- 1.5. Provide regular, independent psychological support for researchers to help them develop coping strategies and identify potential triggers. YES!!!!

Commented [A36]: Editor: Reviewer #1 said: Addressing how individuals without a research team network to rely on can apply these in their situation could further assist those facing challenges with emotionally demanding research.

If you feel this is something you can/want to speak to, I'd encourage you to include it.

Commented [A37]: Editor: If you can link to an example that's great.

Commented [A38]: Editor: Changes made by Ref #2: These are valid points – if you agree, you could rephrase the bullet point as: Offer access to free training and resources [...] Renumerate training or provide time within working hours to ...

Commented [A39]: Editor: Changes made by Ref #2: You can adopt or discard this

Commented [A40]: Reviewer #2: Change in language is confusing- now you are modelling how to say it rather than describing the item.

1.6. Advocate for the importance of considering the mental health and wellbeing of researchers who work on EDR at an institutional level, and implementing relevant practices to support them (e.g., clinical supervision). We believe this ought to form part of every academic institution's Code of Practice for Research. Don't forget the timelines issue...what about sick leaves? Is there ecomensation for students who are hired as assistants if they are then made unwell by the work? In my country there is a whole administrative and budget pice that would need to be placed out in the open for transparent discussion.

Commented [A41]: Editor: Changes made by Ref #2:
There are some good points here – it is at your discretion to add some or not.

2. *Work patterns and environment*

- 2.1. Organise suitable working arrangements that provide work spaces for researchers to work alongside one another for regular check-ins and support (e.g., being able to express certain thoughts and feelings, and share them with someone who understands).
- 2.2. Discourage unhealthy working practices, such as reading potentially distressing material late into the day, at night, or in private spaces. Again, you do not know everyone's situation, maybe there are people who find it safer psychologically to do these things - you need to establish why, as a rule, they are not a good idea in relations to "EDR"
- 2.3. Limit exposure to potentially distressing material by allocating researchers to other project tasks for at least 50% of their working week. YES!!!
- 2.4. Arrange regular team meetings in order to give researchers access to supervision, and encourage reflection and discussion among the research team. YES!!!

3. *Knowing yourself and your limits*

- 3.1. Encourage reflection and discussion among the research team and the researchers about what may be triggering or difficult to work on. When working on EDR, it is important to develop a good understanding of this, as it will likely vary for different people. Being open and transparent with colleagues about what you feel that you can and cannot work on will allow the team to find ways to best support you (e.g., by allocating certain tasks to other team members). This ultimately helps to protect researchers and their mental health and wellbeing, while ensuring that the research project receives the very best of them and their expertise.
- 3.2. Encourage researchers to take regular breaks and switch tasks, and facilitate this by building allowances into their schedules.
- 3.3. Accept and respect if working with EDR is not/does not feel right for someone. There are many reasons why someone may not wish to embark on or join an EDR project, and there are just as many reasons why someone may wish to leave an EDR project. If someone decides it is just not for them, they do not need to provide an explanation or justification to anyone; it is by no means a reflection of them as a person or a professional. LOVELY conclusion

References

- 1) Kumar, S., & Cavallaro, L. (2018). Researcher Self-Care in Emotionally Demanding Research: A Proposed Conceptual Framework. *Qualitative Health Research*, 28(4), 648-658.
- 2) Silverio, S., Sheen, K., Bramante, A., Knighting, K., Koops, T., Montgomery, E., . . . Sandall, J. (2022). Sensitive, Challenging, and Difficult Topics: Experiences

and Practical Considerations for Qualitative Researchers. *International Journal of Qualitative Methods*, 21, 160940692211247.